# Parallel Rendezvous Strategy for Node Association in Wi-SUN FAN Networks

**DOI:** 10.3390/s25196213

**Published:** 2025-10-07

**Authors:** Ananias Ambrosio Quispe, Rodrigo Jardim Riella, Luciana Michelotto Iantorno, Patryk Henrique da Fonseca, Vitalio Alfonso Reguera, Evelio Martin Garcia Fernandez

**Affiliations:** 1Department of Electrical Engineering, Federal University of Paraná-UFPR, Curitiba 81531-980, Brazil; riella@lactec.com.br (R.J.R.); evelio@ufpr.br (E.M.G.F.); 2Future Grid Competence Center, Lactec Institutes, Curitiba 80215-090, Brazil; luciana.iantorno@lactec.com.br (L.M.I.); patryk.fonseca@lactec.com.br (P.H.d.F.); 3Postgraduate Program in Energy Systems, Federal Technological University of Paraná-UTFPR, Curitiba 80230-901, Brazil; 4Department of Information Technologies, Universidad Tecnologica del Uruguay-UTEC, Rivera 40000, Uruguay; vitalio.alfonso@utec.edu.uy

**Keywords:** parallel rendezvous, Wi-SUN FAN, LLN, trickle timer

## Abstract

The Wi-SUN FAN (Wireless Smart Ubiquitous Network Field Area Network) standard facilitates large-scale connectivity among smart devices in utility networks and smart cities. Specifically designed for Low-Power and Lossy Networks (LLNs), Wi-SUN FAN supports the formation of multiple Personal Area Networks (PANs) and mesh topologies with multi-hop transmissions. However, the node association process, divided into five junction states, often results in prolonged connection times, particularly in multi-hop networks, thereby limiting network scalability and reliability. This study analyzes the factors affecting these delays, with a particular focus on Join State 1 (JS1), which relies on PAN Advertisement (PA) packets that use asynchronous communication and the trickle timer algorithm, frequently causing significant delays. To overcome this challenge in JS1, we propose the Parallel Rendezvous (PR) strategy, which forms synchronized clusters of unassociated nodes and leverages the standard’s PAN Advertisement Solicit (PAS) packets to rapidly disseminate network information. The proposed algorithm, PR Wi-SUN FAN, is evaluated through simulations in various network topologies, demonstrating notable improvements in linear, fully connected, and mesh scenarios. The most significant gains are observed in the linear topology, with reductions of up to 71.22% in association time and 59.56% in energy consumption during JS1.

## 1. Introduction

With the development of the Internet of Things (IoT), various standards have been adopted to address the large-scale demand for interconnecting smart devices in applications such as smart cities and utility services (electricity, water, gas, etc.). The need for low energy consumption, low-cost devices, and reduced data traffic has driven the development of standards such as Sigfox, Zigbee, Lora, and Wi-SUN (Wireless Smart Ubiquitous Network) alliances [1].

This study focuses on the Wi-SUN FAN profile, a standard within the Wi-SUN alliance designed for LLNs that require interoperable solutions [2,3] in various applications, particularly utility networks and smart cities [4]. In these contexts, thousands of nodes are interconnected [5,6], forming multiple PANs. The Wi-SUN FAN standard comprises open protocols, primarily at the physical and data link layers compatible with IEEE 802.15.4 [7]. It features a mesh-type topology, enabling multi-hop data transmission via the Routing Protocol for Low-Power and Lossy Networks (RPL) [8], which enables the formation of dynamic networks, supporting data rates of 50 kbps and 150 kbps in version 1.0, and up to 2.4 Mbps in version 1.1 [7,9]. The standard allows coexistence and simultaneous operation with Frequency Shift Keying (FSK) and Orthogonal Frequency Division Multiplexing (OFDM) modulations [9,10], frequency hopping for interference avoidance, and interoperability among different manufacturers.

As a mesh network, Wi-SUN FAN employs a hierarchical solution with three node types: Border Router (BR), Router (R), and Leaf (L). The network must support communication in dense environments having high data flow, while accommodating devices with limited resources. This necessitates special attention to node connecting procedures, which include discovery, joining, authentication, and routing configuration.

According to the Wi-SUN FAN standard [11], this process comprises five join states (JSs):JS1: PAN discovery and selection;JS2: Authentication;JS3: PAN configuration;JS4: Routing configuration;JS5: Operational state.

Transitioning through all these states by the node significantly delays overall network formation, a problem identified by Junjalearnvong et al. [12], who reported slow connection times in multi-hop networks. These times exceed 5 min per node in a linear network, as experimentally verified in [12,13]. Although Wi-SUN FAN documentation provides parameter recommendations for small-scale (tens/hundreds of nodes) and large-scale (thousands of nodes) networks [14], empirical studies across diverse topologies—including point-to-point [15], multi-hop, and mesh deployments [12,13,16,17]—consistently report high connection times. For example, first-node connection times surpass 3 min in small-scale networks and 15 min in large-scale deployments [16].

When the BR must be restarted, grid recovery may be prolonged. Adverse conditions, such as operational failures or extreme weather, can compromise electrical distribution systems, causing interruptions in smart networks. After power restoration, network recovery remains slow due to protocol limitations—Silicon Labs tests shown that a 100-node network requires over an hour to restore [16]. These delays hinder the network’s primary purpose: connecting and enabling communication between a large number of devices.

To address the node connection time problem, it is important to identify and differentiate the processes in each JS. In Wi-SUN FAN, the node connection time is determined as the total duration of the transient states (JS1–JS4). In contrast, JS5 represents the operational state, where the node is fully associated and functional.

Several studies have sought to optimize connection times by focusing on specific states. In [12], the authors address node connection time by analyzing the trickle timer algorithm [18] in JS1 and JS3 in a multi-hop network. This algorithm is used to control asynchronous packet transmission in these states. They propose a parameter configuration scheme that controls the transmission rates of discovery and configuration packets through dynamic adjustment of message-sending periods.

Implementing a trickle timer configuration scheme for JS1 and JS3 packet transmission reveals a trade-off among connection time, latency, and network scalability [19]. Longer transmission intervals increase connection times but improve latency and scalability. Shorter intervals reduce connection times but degrade scalability and increase latency due to network management overhead [16].

During JS2 authentication, security parameters (authentication data and keys) are established and verified for new nodes joining the network [14]. To reduce connection time, the standard implements security credential storage, enabling nodes to skip JS2 after reboots by reusing prior authentication.

For JS4 (network routing), RPL protocol optimizations target connection time reduction through node load balancing [20,21], enhanced network stability [22,23], accelerated reconnections [24], node mobility [25,26], and trickle timer configuration for RPL packet control [27,28,29]. Channel hopping schedules—governing listen/transmit durations for unicast and broadcast communications—also impact connection processes. Studies confirm that modifying these durations, particularly in JS4, significantly affects performance [30,31].

Despite these efforts, further alternatives are needed to reduce network formation times. To this end, we propose PR Wi-SUN FAN, a cooperative association method based on a Parallel Rendezvous (PR) strategy. This strategy, previously applied in cognitive radio [32,33] and Time-Slotted Channel Hopping (TSCH) networks [34,35], can enable rapid dissemination of PAN Advertisement (PA) packets within a cluster. To the best of our knowledge, and based on the literature review, the use of clusters for unassociated nodes in the initial association process has not yet been explored in Wi-SUN FAN.

In Wi-SUN FAN, unassociated nodes actively transmit PAN Advertisement Solicit (PAS) packets for network discovery [14,36]. PR Wi-SUN FAN leverages these PAS packets to establish synchronized clusters during non-JS1 channel scanning, where nodes acquire minimal association parameters that facilitate the discovery of channel hopping schedules within the cluster. This mechanism accelerates the dissemination of PA packets within clusters, thereby accelerating JS1 processing and reducing energy consumption. To achieve this, the proposed method is implemented in the Contiki-NG operating system, extending the work of [17]. Furthermore, the evaluation of the proposed method, including the entire process of connecting nodes, demonstrates that using PR contributes to reducing the time required to form the network. Our main contributions are as follows:Stochastic modeling of association times, deriving expectations for node association in linear, fully connected, and mesh Wi-SUN FAN networks.Design of a cooperative association method based on PAS packets to form synchronized clusters in JS1 (PR Wi-SUN FAN), supported by stochastic modeling, demonstrating how Parallel Rendezvous reduces association time during JS1 in Wi-SUN FAN.Implementation, simulation, and evaluation of the Parallel Rendezvous method on Wi-SUN FAN in JS1 using Contiki-NG, showing that the proposed approach reduces both association time and energy consumption.

Next, Section 2 provides the background of the Wi-SUN FAN standard, while Section 3 presents a theoretical explanation of JS1 association. Section 4 introduces PR Wi-SUN FAN and analyzes its behavior during the JS1 association process. Section 5 describes the configurations of the studied scenarios. Section 6 presents the results. Finally, Section 7 concludes the paper.

## 2. Wi-SUN FAN: Background and Overview

Wi-SUN FAN is a standard based on the IEEE 2857-2021 [14] specification, which defines protocols for the first four layers of the OSI model. The physical and data link layers are implemented using IEEE 802.15.4 [7]; the network layer utilizes IPv6, ICMPv6, RPL, and 6LoWPAN; and the transport layer supports both UDP and TCP. Security mechanisms are based on IEEE 802.1X, IEEE 802.11i, and EAP-TLS protocols.

The Wi-SUN FAN network has a hierarchical topology. It is formed based on a directed acyclic graph (DAG) rooted at the BR, which acts as the main parent for nodes R and L. To join the network, nodes R and L go through five joining states before reaching the operational state, a procedure referred to as the node connection process. This work focuses specifically on JS1, as it is in this phase that the implementation of the described optimization proposal occurs. JS1 corresponds to the discovery and selection process, which follows a scheduling program for channel management, explained below.

### 2.1. Wi-SUN FAN Scheduling

In Wi-SUN FAN networks, channel hopping scheduling methods are defined for both unicast and broadcast transmissions. This study focuses exclusively on unicast scheduling. In this scheme, time is divided into discrete slots, and the duration of a Unicast Slot (US) is termed the Unicast Dwell Interval (UDI). According to the standard, the UDI length is defined between 15 ms and 255 ms [14]. Channel assignments are generated based on each node’s EUI-64 address, resulting in unique channel sequences for every node. Figure 1 illustrates an example of unicast channel hopping for three nodes, with four available channels (CH1–CH4).

A channel function defines a pseudo-random sequence used for channel hopping scheduling. It establishes a method for determining, from the list of available physical channels, the specific channel on which a node operates at any given time. Several channel functions are supported at the link layer for unicast operation, including fixed channel, TR51CH, DH1CF, and vendor-defined.

Since each node generates its own unicast schedule, transmitting a frame to a neighbor requires calculating the frequency channel on which the neighbor is currently listening. The transmission must then occur on that channel. To determine the current position in a node’s channel hopping sequence, both the transmitter and receiver must use the same predefined function, which relies on the node’s EUI-64 address, channel number, and Unicast Slot (US) number. To perform the discovery process over this channel schedule, Wi-SUN FAN uses an active scanning method.

### 2.2. Active Scan in Wi-SUN FAN

Wi-SUN FAN networks utilize an active scanning method for node discovery and association. In this mode, a node seeking to join a network initiates the process by transmitting PAS packets to request and accelerate PA transmissions from advertising nodes within the PAN. This contrasts with passive scanning, where a node scans channels without sending requests until it receives a PA packet. For node R, passive scanning is consistently performed following a transmission, whereas node L enters a sleep mode to conserve energy.

The active scanning mechanism involves sending multiple PAS packets sequentially across each available physical channel (CH1, CH2, …, CH*n*)—termed PAS_train in this work. Upon receiving a PAS packet, an advertising node accelerates PA transmissions, also following a sequential channel order (denoted PA_train). Figure 2 illustrates this active scanning process alongside a passive scanning comparison.

During the discovery and joining process in JS1, Wi-SUN FAN employs the MLME-WS-ASYNC-FRAME service [14], a primitive mechanism that manages asynchronous transmission of PA and PAS trains. Figure 3 illustrates these transmissions, which must span all available channels. A PA_train or PAS_train is characterized by the packet duration $T_{c}$, the interval between consecutive packet transmissions $T_{e}$, and the total train duration, expressed as $C\cdot T_{e}$, where *C* denotes the number of available channels.

The MLME-WS-ASYNC-FRAME mechanism triggers PA or PAS packet transmission when the trickle timer requires a transmission.

### 2.3. Application of Trickle Timer on Wi-SUN FAN

A Wi-SUN FAN junction node must maintain the trickle timer to control the transmission frequency of PAS and PA packets in the JS1 and JS5 processes, respectively. The operation of the trickle timer is described in the RFC6206 documentation [18], where the algorithm is configured with three parameters: $I_{\min}$, the minimum interval, defined in time units such as milliseconds or seconds; $I_{\max}$, the maximum interval, calculated by successively doubling $I_{\min}$ a specified number of times; and *k*, the redundancy constant, which limits the number of allowed transmissions to avoid redundant messages.

Figure 4 illustrates the initial behavior of the trickle timer, which starts by resetting the consistency counter *c* to zero. Next, the interval length *I* is randomly selected within the range $\left[I_{\min},I_{\max}\right]$, and the transmission time *t* is chosen uniformly within the subinterval [I/2,I] [37]. In the context of Wi-SUN FAN, time *t* denotes the scheduled start time of the PA and PAS trains, as detailed in Section 2.2. When a node receives identical information from its neighbors, it increments the counter *c*. Transmission at time *t* only occurs if *c* is less than the redundancy constant *k*; otherwise, the transmission is suppressed. At the end of each interval *I*, the algorithm doubles its duration up to the maximum interval defined by $I_{\max}$. If the current interval has not yet expired, *c* is reset to zero and *t* is randomly selected from the range [I/2,I]. A relevant aspect of the trickle timer’s operation is the listen-only period, which corresponds to the first half of each interval. This structure divides the interval into two different phases: a listen-only phase and a transmit phase during which transmissions can occur. The behavior of the trickle timer is described in six steps in the RFC6206 documentation [18].

In addition, the trickle timer uses the concepts of consistent, inconsistent, and generic events to suppress unnecessary messages and to adapt the scheduling of maintenance packets. This enables the trickle timer to be adapted and adopted by various protocols that interpret these events according to their specific semantics. In Wi-SUN FAN, the transmission of PAS and PA packets incorporates these considerations, as described in the standard documentation [14], which relate to starting, restarting, or stopping the timer.

The Wi-SUN FAN standard also provides recommendations for trickle timer configuration according to network size, differentiating between small-scale networks with tens of nodes per PAN ($I_{\min}=15$ s and $I_{\max}=60$ s), which use shorter intervals to prioritize responsiveness, and large-scale networks with thousands of nodes per PAN ($I_{\min}=60$ s and $I_{\max}=16$ min), which use longer intervals to minimize congestion and optimize scalability [14].

## 3. Stochastic Modeling of JS1-Based Discovery in Wi-SUN FAN

In Wi-SUN FAN networks, a node in JS1 initiates discovery by performing a channel scan (CHscan) to detect PA packets across available channels. This scan follows a sequential order defined by the channel array *V*, which is calculated using a channel function in a round-robin manner. For this study, we implement an approach where the connection process transitions directly from JS1 to JS5 under version 1.0 of the standard. By bypassing intermediate joining states, nodes operate exclusively within JS1, enabling focused analysis of this specific phase.

Algorithm 1 describes the Wi-SUN FAN joining process for nodes in JS1. The process begins by defining the number of channels (*C*), essential for channel hopping (line 1). A channel function then generates a pseudo-random sequence based on the node’s EUI-64 address, stored in array V[] (line 2). The channel index ($I_{c}$) selects the current channel within V[] (line 3), while the Unicast Dwell Interval (UDI) determines the listening duration per channel before switching (line 4). $I_{c}$ is initialized to 0 (line 5), initiating the scan at the first channel in V[]. The node enters JS1 (line 6), the initial connection state. Here, the trickle timer for PAS packets activates (line 7), and the node scans the channel $V[I_{c}]$ (line 8), following the generated sequence.

After initial configuration, the node enters a continuous loop during network connection (line 9). The loop first checks for UDI timer expiration; if expired, it restarts the UDI timer, increments $I_{c}$, and advances to the next channel in V[] (lines 10–14). While unassociated and in JS1, if the trickle timer permits PAS transmission, the node sends PAS to request PA packets (lines 15–17). Upon receiving PA, it stops the PAS trickle timer and transitions to JS5 (lines 18–22). In JS5 (associated state), the node starts the PA trickle timer to announce its presence to its neighbors, facilitating network formation (lines 23–26). Throughout, it continuously monitors its join state and scans channels per V[] (line 27).
**Algorithm 1** Wi-SUN FAN discovery process on JS11:*C*: number of available channels2:V[]← channel_function()3:$I_{c}$: channel index4:UDI: unicast dwell interval5:$I_{c}$← 06:node = JS17:start of PAS trickle timer8:$CHscan\leftarrow V[I_{c}]$9:**while** node is in the process of connecting **do**10:    **if** UDI timer has expired **then**11:        resets the scan time12:        $I_{c}\leftarrow \left(I_{c}+1\right)$13:        $CHscan\leftarrow V[I_{c}]$14:    **end if**15:    **if** node = JS1 **then**16:        node is not associated17:        transmits PAS when the trickle timer allows18:        **if** PA is received **then**19:           stop of PAS trickle timer20:           node ← JS521:        **end if**22:    **end if**23:    **if** node = JS5 **then**24:        node is associated25:        start of PA trickle timer26:    **end if**27:**end while**

The Extensive Authentication Protocol over LAN (EAPoL) candidate list was not implemented in JS1. This EAPoL candidate list is used to determine the PAN to which the node will attempt to connect, based on the lowest route cost, which mainly depends on information obtained in JS4. Therefore, the node selects only the first PA received to advance to JS5. This choice aims to reduce the complexity of the implementation and allow for a more controlled study by isolating JS1.

Based on these considerations, a theoretical analysis will be carried out following the modeling presented in [34], considering three network topologies—linear, fully connected, and mesh—as shown in Figure 5.

### 3.1. Linear Network in JS1—Wi-SUN FAN

Figure 6 depicts a packet exchange and association process in a linear network topology comprising a BR (node 0, initially associated) and N=3 R nodes (nodes 1–3) attempting to join. This worst-case scenario (Figure 5a) has a network diameter H=N hops and *C* available channels. Node 0 is connected at $t_{0}$. During association, node *j* (j=1 to *N*) listens to PA packets from node j−1 and PAS packets from non-associated node j+1, except node *N*, which only hears PA packets from node N−1. Association requires sequential dependency: node *j* connects only after node j−1. Figure 6 shows this timeline, where node *N*’s connection depends on node N−1’s prior association.

For each node, a sequence of channels is shown as dotted horizontal bars, with the sequence length determined by the number of channels (*C*). Transmissions of PA_train and PAS_train are represented by gray and orange vertical bars, respectively, while receptions of PA and PAS packets are indicated by green and blue vertical bars, respectively.

In this context, node 0 (distinct from other nodes) initiates network formation at $t_{0}=0$. To measure the linear network’s formation time, the time offsets ($\Delta {\mathrm{offset}}_{j}$) of each router node are disregarded. This alignment ensures all measurements are referenced to the Border Router’s start time.

An important point is the disregard, in our calculation, of the reception of PAS packets at time $tr_{j}$ for the case of an associated node, since this does not affect the start of the time of the first PA_train at time $tp_{j}$. This situation generates an inconsistency in the trickle timer that resets the interval for $I_{\min}$. However, this has no impact, since the interval already uses $I_{\min}$. Furthermore, in the case of PAS reception in non-associated nodes, we can also disregard the generation of a consistency in the trickle timer, which suppresses the sending of the PAS_train, which is not considered for our calculation.

The time required for node *j* to associate with the Wi-SUN FAN network is represented by $ta_{j}$, which is the time elapsed from the moment node j−1 associates until the moment node *j* hears a PA from node j−1. This can be represented as(1)$$\Delta ta_{j}=ta_{j}-ta_{j-1},\;0<j\leq N.$$

Since $t_{0}=0$, the value of $ta_{0}$ is also 0, resulting in $\Delta ta_{1}=ta_{1}$. This corresponds to the time interval between the association of node 0 and the reception of the PA by node 1, and it is therefore equal to the association time of node 1. The instant $ta_{1}$ can be expressed as the sum of the start of the PA_train, denoted by $tp_{0}$, and the delay $T_{d}$ until node 1 receives one of the PA_train packets:(2)$$ta_{1}=tp_{0}+T_{d}.$$

The value of $tp_{0}$ is determined by the interval specified in the trickle timer, which follows a continuous uniform distribution [37] over the interval [I/2,I]. Therefore, the expected value of the start of PA_train is $E\left[tp_{0}\right]=\frac{3I}{4}$.

Let $T_{d}$ be the time between the start of the PA_train and the instant at which a PA packet is received by node 1, assuming that at least one packet will be received, in a guaranteed reception scenario occurring at most within $C\cdot T_{e}$ time units, where *C* denotes the number of available channels, and $T_{e}$ is the interval between consecutive packets of the PA_train. Assuming that the start of the hop of node 1 is uniformly distributed over the cycle of *C* channel, then $T_{d}∼U\left[0,C\cdot T_{e}\right)$ with $E\left[T_{d}\right]=\frac{C\cdot T_{e}}{2}$. Consequently, the expected value of $ta_{1}$ is given by(3)$$E\left[ta_{1}\right]=E\left[tp_{0}\right]+E\left[T_{d}\right]=\frac{3I}{4}+\frac{C\cdot T_{e}}{2}.$$

Considering that node j−1 does not start sending a PA_train until it has joined the network, this implies that the elapsed time between a node joining and the PA being sent to the next node is the same for all nodes in the network. This assumption is reasonable because, in the context of the Wi-SUN FAN algorithm, the behavior of nodes is sequential in a linear network. Therefore, the expected value of the time between node association, denoted as $\Delta ta_{j}$ in this analysis, can be assumed to be the same for all nodes in the network, i.e.,:(4)$$E\left[\Delta ta_{j}\right]=E\left[\Delta ta_{i}\right],\;\forall i,j\in [1,N].$$

Using (Equation 4) and rewriting (Equation 1) in a convenient form, we have(5)$$E\left[ta_{j}\right]=E\left[ta_{j-1}+\Delta ta_{1}\right].$$

From this, by solving a linear recurrence equation, the expected value of the time to form a linear network of diameter H=N, denoted as $ta_{N}$, can be expressed as(6)$$E\left[ta_{N}\right]=N\cdot E\left[ta_{1}\right].$$

### 3.2. Fully-Connected Network in JS1—Wi-SUN FAN

In a fully connected network, all nodes can transmit and receive directly to and from any other nodes in the network. This means that there are no restrictions on direct communication between nodes, allowing for full connectivity, as shown in Figure 5b. In this scenario, when an unassociated node attempts to join the network, it can join through any of its neighbors that are already joined.

Let $ta_{ji}$ be the time required for node *j* to hear a PA packet sent by node *i*. Based on this definition, the association time $ta_{j}$ in a network with *N* nodes trying to associate can be expressed as(7)$$ta_{j}=\min\left(\left\{X_{ji}\right\}\right),\;\forall i\neq j,$$
where $\min(\left\{X_{ji}\right\})$ indicates that node *j* selects the shortest PA packet reception time among all nodes *i*. Now, considering the last node to join the network, 0<l≤N, from (Equation 7), the expected time to form a network of *N* unassociated nodes into a fully connected network can be expressed by the association time of the last node:(8)$$E\left[ta_{N}\right]=E\left[ta_{l}\right]=E\left[\min(\left\{X_{li}\right\})\right],\;\forall i\neq l.$$

Considering an ideal scenario and ignoring potential issues that could interfere with the reception of PAs by the nodes, it is possible to establish a maximum limit for the time node *l* would take to receive a PA from the BR ($X_{l0}$). This maximum limit is determined by the trickle timer and the PA_train. Assuming that this limit exists, the maximum time for node *l* to receive a PA from the BR is defined as $T_{M}$, and we have $E[ta_{N}]$ ≤ $T_{M}$, where $T_{M}=I+\left(C\cdot T_{e}\right)$, with *I* being the interval defined by the trickle timer, *C* the maximum number of available channels, and $T_{e}$ the interval time between PAs in the PA_train.

In a fully connected network, unlike in a linear network, the expected time to form the network, $E[ta_{N}]$, does not increase linearly with the number of nodes; instead, the expected value decreases as the number of nodes increases. That is, the value calculated by Equation (Equation 8) tends to decrease as the number of elements in $\left\{X_{li}\right\}$ increases. This behavior is discussed in Section 4 in the context of the PR Wi-SUN FAN fully connected network.

### 3.3. Mesh Network in JS1—Wi-SUN FAN

In a mesh network, illustrated in Figure 5c, nodes can have more than two neighbors, meaning that each node in the network can be connected to multiple other nodes. However, it is important to note that despite having multiple neighbors, a node in a mesh network can neither transmit to nor receive from any node other than its direct neighbors. For example, in Figure 5c, node 3 can only associate with the network through nodes 1 and 4, depending on which node hears a PA first.

For a node in a mesh network, let $\left\{D_{ji}\right\}$ be the set containing the specific nodes that are within the direct communication range of that node and can send PAs to it. Given this set, (Equation 7) can be rewritten as(9)$$ta_{j}=\min\left(\left\{X_{ji}\right\}\right),\;\forall i\in \left\{D_{ji}\right\},$$
and the expected time to connect the entire network can be expressed as(10)$$E\left[ta_{N}\right]=E\left[ta_{l}\right]=E\left[\min(\left\{X_{li}\right\})\right],\;\forall i\in \left\{D_{ji}\right\}.$$

Since the number of elements in $D_{ji}$ is smaller than *N*, the expected time to form a mesh network is greater than that of a fully connected network with the same number of nodes. Furthermore, the network diameter impacts the result of Equation (Equation 10), but since the diameter *H* is smaller than the number of nodes *N*, it becomes evident that $E[ta_{N}]$ is upper-bounded by Equation (Equation 6). That is, we should expect $E[ta_{N}]$ for a mesh network to lie somewhere between the values obtained for a fully connected network and a linear network. This intuitive interpretation is corroborated by the results presented in Section 6.

## 4. PR Wi-SUN FAN: Parallel Rendezvous for Wi-SUN FAN
Association

Our proposal introduces a cooperative joining scheme, based on the Parallel Rendezvous strategy, to accelerate the connection process of type R nodes in the Wi-SUN FAN standard. The idea is to take advantage of the PAS packets transmitted by non-associated nodes to form synchronized clusters during the channel scanning performed by these same nodes. Each cluster generates, for each non-associated node, a list containing information about its non-associated neighbors (PR_table), including the unicast channel hopping time scheme. In this way, when a member node of the cluster manages to connect to the Wi-SUN FAN network, it quickly disseminates PA packets to its list of neighbors through unicast communication; this transmission is not governed by the trickle timer. After completing the dissemination within its cluster, the node can continue with asynchronous PA transmission, now governed by the trickle timer. Figure 7 illustrates the formation of two clusters in a PR Wi-SUN FAN network.

The cluster operates according to the following considerations:The unassociated node operates in a fully distributed manner, independently generating its own PR_table and making decisions based on information received from neighboring nodes, without reliance on a central controller or server.The unassociated node sorts its PR_table according to the signal strength of the PAS packet received from its neighbors, which must be at an acceptable level. This order is maintained when sending a PA packet using unicast communication (PA_unicast) to each neighbor in the list.The unassociated node can remove a neighbor from its PR_table if it receives a packet from the PA_train, indicating that the neighbor in no longer in SJ1.

### 4.1. The PR Wi-SUN FAN Discovery Process on JS1

Algorithm 2 describes the PR Wi-SUN FAN process, which follows the Parallel Rendezvous strategy and allows unassociated nodes to exchange synchronization information using PAS packets during JS1. From lines 1 to 21, the procedure is the same as that described in Algorithm 1. If an unassociated node receives a PAS packet, it stores the necessary synchronization information in a table called PR_table (lines 22–24 in Algorithm 2). Once the node has joined in JS5 (lines 26–28 in Algorithm 2), the PR Wi-SUN FAN node sends a PA packet via unicast to each of its unassociated neighbors registered in the PR_table (lines 29–31 in Algorithm 2).

In Algorithm 2, the same considerations as in Algorithm 1 are followed, and the EAPoL candidate list is not implemented. This choice also aims to reduce the complexity of the implementation, enabling a more controlled study by isolating JS1. The analyses are performed on linear, fully connected, and mesh topologies, as shown in Figure 5.
**Algorithm 2** PR Wi-SUN FAN discovery process on JS11:*C*: number of available channels2:V[]← channel_function()3:$I_{c}$: channel index4:UDI: unicast dwell interval5:$I_{c}$← 06:node = JS17:start of PAS trickle timer8:$CHscan\leftarrow V[I_{c}]$9:**while** node is in the process of connecting **do**10:    **if** UDI time has expired **then**11:        resets the scan time12:        $I_{c}\leftarrow \left(I_{c}+1\right)$13:        $CHscan\leftarrow V[I_{c}]$14:    **end if**15:    **if** node = JS1 **then**16:        node is not associated17:        transmits PAS when the trickle timer allows18:        **if** PA is received **then**19:           stop of PAS trickle timer20:           node ← JS521:        **end if**22:        **if** PAS is received **then**23:           *PR_table* ← unicast scheduling data24:        **end if**25:    **end if**26:    **if** node = JS5 **then**27:        node is associated28:        start of PA trickle timer29:        **for** each node in *PR_table* **do**30:           predict and send *PA_unicast* to the node31:        **end for**32:    **end if**33:**end while**

#### 4.1.1. Linear Network in JS1—PR Wi-SUN FAN

Figure 8 shows an example of a node association process for a topology consisting of a BR (node 0), which starts already associated, and *N* unassociated R nodes in the network (nodes 1 to 3), which are trying to associate. For each R node, a sequence of channels is shown, represented by dotted horizontal bars, which depends on the number of available channels, *C*. The transmissions of the PA_train, PAS_train, and PA_unicast are represented by vertical bars in gray, orange, and red, respectively, while the receptions of PA and PA_unicast packets are indicated by green vertical bars, and the receptions of PAS packets are shown in blue.

In this scenario, in the general case, after node j−1 joins the network, 0<j≤N, and it can immediately send PA_unicast packets to node *j* if it has already received a PAS packet and stored it in its PR_table for node *j*. Otherwise, node *j* will have to wait for a period determined by its trickle timer, which corresponds to the conventional Wi-SUN FAN process. Considering that the transmission time of PAs to neighboring nodes in the Parallel Rendezvous scenario is negligible compared to the time required for these neighbors to complete their association, the expected value of the association time increment for node *j* can be expressed as(11)$$E\left[\Delta ta_{j}\right]=E\left[\Delta ta^{n\mathrm{PAS}}\right]\left(1-P_{j}\right),\;0<j\leq N,$$
where $\Delta ta^{n\mathrm{PAS}}$ is the expected time to associate without a previously received PAS, and $P_{j}$ is the probability that node j−1 has already received a PAS from node *j* before associating. In other words, this extra time will only exist if node j−1 has not received a PAS from node *j*.

Since the BR (node 0) has been associated with the network since $t_{0}$, it does not have a PR_table and will not store information about PAS packets. Therefore, $P_{1}=0$. Consequently, the association time for node 1 will always be as if no PAS had been received. For this case, we have(12)$$E\left[\Delta ta_{1}\right]=E\left[\Delta ta^{n\mathrm{PAS}}\right]=E\left[ta_{1}\right].$$

In general, the total time for node *j* to join is the sum of the joining time of the previous node and the additional time required for node *j* to join, i.e.,(13)$$E\left[ta_{j}\right]=E\left[ta_{j-1}\right]+E\left[\Delta ta_{j}\right].$$

Then, substituting (Equation 11) and (Equation 12) into (Equation 13), we obtain(14)$$E\left[ta_{j}\right]=E\left[ta_{j-1}\right]+E\left[ta_{1}\right]\left(1-P_{j}\right).$$

Assuming that the time during which node j−1 can hear a PAS from node *j*, denoted by $X_{j-1,j}$, follows the trickle timer schedule and occurs within distinct time windows, we simplify our theoretical analysis by modeling $X_{j-1,j}$ as a uniformly distributed random variable over the interval $\left[0,T_{M}\right]$. Based on this assumption, the probability that node j−1 has received a PAS from node *j* before associating, at the average time $E\left[ta_{j-1}\right]$, is given by(15)$$P_{j}=\left\{\begin{array}{cc} \frac{\min\left(E\left[ta_{j-1}\right],T_{M}\right)}{T_{M}}, & j=2,3,\ldots ,N,\\ 0, & j=1. \end{array}\right.$$

The expected time for all *N* R nodes to associate with the network is obtained by substituting (Equation 15) into (Equation 14), resulting in(16)$$E\left[ta_{N}\right]=T_{M}\left[1-{\left(1-\frac{E\left[ta_{1}\right]}{T_{M}}\right)}^{N}\right].$$

Equation (Equation 16) shows an inverse exponential relationship, suggesting a cumulative gain as the probability of early listening increases. If the time $E[ta_{1}]$ is small relative to $T_{M}$, nodes are more likely to hear PASs earlier, and the network forms more quickly.

#### 4.1.2. Fully Connected Network in JS1—PR Wi-SUN FAN

When the PR Wi-SUN FAN algorithm is used in a fully connected network, as illustrated in Figure 5b, the first node to join the network sends PA packets to all its neighbors with which it can communicate. Subsequent nodes do the same, creating a kind of “domino effect.” This means that as more nodes join the network, the chances of receiving a PA increase, since each node can act as a synchronization point for the others. Therefore, as discussed earlier, PR Wi-SUN FAN is expected to perform better than the conventional Wi-SUN FAN in this type of network.

In this scenario, as in the previous analyses, we also ignore failures that could prevent nodes from correctly receiving PA and PAS packets. We assume that the time required for a node to announce the presence of the network to its neighbors in its PR_table is negligible compared to $ta_{N}$. Furthermore, for the last node to join the network, denoted as node *l*, we assume that (i) the set of times at which a PA is received from any node i≠l, $\left\{X_{li}\right\}$, consists of independent and identically distributed random variables; and (ii) these variables follow a uniform distribution over the interval $\left[\frac{T_{M}}{2},T_{M}\right]$. At the beginning of the PA_train, it is assumed that the scanning of the channels is uniformly distributed over time, and that the PAs are transmitted on any of the *C* channels with equal probability. Then, the cumulative distribution function of $ta_{l}$ is given by(17)$$F_{ta_{l}}\left(x\right)=P\left(ta_{l}\leq x\right)=1-\prod_{i\neq l}P\left(X_{li}>x\right),$$
where(18)$$P\left(X_{li}>x\right)=\left(\frac{T_{M}-x}{\frac{T_{M}}{2}}\cdot \left(1-\frac{1}{C}\right)\right),\;x\in \left[\frac{T_{M}}{2},T_{M}\right].$$

Since the network consists of *N* nodes, it follows that(19)$$\prod_{i\neq l}P\left(X_{li}>x\right)={\left(\frac{T_{M}-x}{\frac{T_{M}}{2}}\cdot \left(1-\frac{1}{C}\right)\right)}^{N},\;x\in \left[\frac{T_{M}}{2},T_{M}\right],$$
and thus, $F_{ta_{l}}\left(x\right)$ can be expressed as(20)$$F_{ta_{l}}\left(x\right)=\left\{\begin{array}{cc} 0 & ,\;x<\frac{T_{M}}{2}\\ 1-{\left(\frac{T_{M}-x}{\frac{T_{M}}{2}}\cdot \left(1-\frac{1}{C}\right)\right)}^{N} & ,\;\frac{T_{M}}{2}\leq x\leq T_{M}\\ 1 & ,\;x>T_{M}. \end{array}\right.,$$
By taking the derivative of (Equation 20), the corresponding probability density function is obtained as(21)$$f_{ta_{l}}\left(x\right)=\frac{2N}{T_{M}}{\left(\frac{T_{M}-x}{\frac{T_{M}}{2}}\cdot \left(1-\frac{1}{C}\right)\right)}^{N-1},\;\frac{T_{M}}{2}\leq x\leq T_{M}.$$

Now, the expected time for the last node to join the network, $t_{a_{N}}$, can be computed as(22)$$E\left[t_{a_{N}}\right]=\int_{\frac{T_{M}}{2}}^{T_{M}}xf_{ta_{l}}\left(x\right)\;dx=\;\frac{2T_{M}}{N+2}+\frac{T_{M}}{2}\cdot \left(\frac{C}{C-1}\right).$$

This result can be interpreted as an estimate of the average time required for the last node to join the network under an optimistic scenario that disregards failures in the reception of PA and PAS packets. It can be observed that $ta_{N}$ decreases as the number of nodes increases, exhibiting an additional average offset below the maximum bound. This suggests that, with more nodes, it becomes more likely that some will quickly discover the network, thereby reducing the average time for the last node to join.

#### 4.1.3. Mesh Network in JS1—PR Wi-SUN FAN

Figure 5c shows an example of a mesh network configuration. In the PR Wi-SUN FAN algorithm, under an optimistic scenario, it is possible for nodes 1 and 2 to receive PAS packets from nodes 3 and 4, respectively. This allows nodes 1 and 2 to store this information in their PR_table and immediately send PA_unicast packets, enabling nodes 3 and 4 to associate more quickly. Thus, the Parallel Rendezvous algorithm facilitates association between nodes that are within range of each other, allowing the information to be stored in the PR_table.

Since a mesh network can exhibit multiple formation paths by combining the characteristics of a linear network (number of hops) and a fully connected network (number of neighbors with mutual visibility), deriving a closed-form analytical expression becomes more complex. Thus, considering the hybrid behavior of the previously studied network models, we can state that the expected value of $ta_{N}$ lies within the range of final association times for linear and fully connected networks, satisfying(23)$$\frac{2T_{M}}{N+2}+\frac{T_{M}}{2}\cdot \left(\frac{C}{C-1}\right)\leq E\left[ta_{N}\right]\leq T_{M}\left[1-{\left(1-\frac{E\left[ta_{1}\right]}{T_{M}}\right)}^{N}\right],$$
where $E\left[ta_{1}\right]$ is the expected formation time for the first node in a linear network—whether using Wi-SUN FAN or PR Wi-SUN FAN—which can be determined from (Equation 3). Experimental evidence supporting these arguments is presented in the subsequent sections.

## 5. Experimental Setup

To evaluate the performance of our proposal, a comparison was made between Wi-SUN FAN and PR Wi-SUN FAN in JS1. The Contiki-NG operating system was used, following the approach described in [17] to implement Algorithms 1 and 2 of our proposal. Using the Cooja tool from Contiki-NG, several simulations were carried out with BR and R nodes, running on the native Cooja mote [17]. A configuration operating in the Sub-GHz band was used, offering 90 channels and a data rate of 50 kbps in the simulator.

To reflect the asynchronous nature of Wi-SUN FAN networks [38], three topologies were generated: linear, fully connected, and mesh. In each topology, multiple simulations were conducted by varying the activation times of the nodes through the “random seed” parameter, thereby replicating the conditions of a fixed utility service network. In all simulations, R nodes start disconnected, meaning they start from JS1. In every scenario, one BR was deployed, while the remaining N nodes were of type R. In the linear topology, 10 R nodes were deployed, each attempting to associate. In the fully connected and mesh topologies, 10 to 50 R nodes were deployed, also attempting to associate. For the PR_table size, a maximum capacity of 50 neighbors was configured.

Channel scanning performance critically depends on the frequency of PA and PAS packet transmissions; however, since the standard does not specify the interval $T_{e}$ for these transmissions and mandates that the UDI length remain within a defined range, multiple configurations were implemented for our selected channel function. This function generates a pseudorandom sequence whose length matches the number of channels, thereby simplifying the analysis. These configurations involve the parameters UDI and $T_{e}$. Table 1 presents the configurations used for different numbers of channels in the implemented topologies.

These configurations ensure that PA or PAS packets are transmitted at least once across the entire pseudo-random sequence, which operates according to a round-robin cyclical schedule.

To configure the trickle timer at the beginning of the PA or PAS packet train transmission, the following parameters were used: $I_{\min}=15$ s, $I_{\max}=60$ s, and k=1, for both Wi-SUN FAN and PR Wi-SUN FAN, in a small-scale configuration [14]. The only variation occurred in the linear topology of PR Wi-SUN FAN, where k=2 was used in order to redundantly transmit PAS packets and prevent them from being quickly suppressed [18]. This configuration aims to increase the effectiveness of storing the information of the sole neighbor in the PR_table.

The performance of Algorithms 1 and 2 is evaluated in terms of the average time required to form a network with N+1 nodes, including the BR and the R nodes, during network formation in JS1. To obtain the data, 100 simulations were conducted for each number of channels. The topologies were constructed using Cooja. Figure 9 illustrates the topologies built for the maximum number of nodes used in this study.

### Energy Consumption

In terms of power consumption, both the BR and R nodes in Wi-SUN FAN keep their radio interfaces continuously on during the association and operation processes. For R nodes specifically, this is necessary to enable communication from other nodes positioned below them, as children. Due to their role within the Wi-SUN FAN network, R nodes are not expected to implement sleep slots, which are a common power-saving mechanism.

To estimate energy consumption, the linear network topology shown in Figure 9a is used. Energy consumption is measured at the end of each R node’s association time ($ta_{j}$). Therefore, the total energy consumption in JS1 of node *j*, $E_{j}$, can be represented as(24)$$E_{j}=ta_{j}\cdot Pa,$$
where Pa is the power consumed by the node with the radio on during the association phases.

BR and R nodes alternate between two modes: transmission (Tx) and reception (Rx). To estimate, for example, the value of Pa, the power consumption in Tx and Rx modes is summed. This is calculated by multiplying the supply voltage by the current consumed in each node. For our analysis in this work, we used the following specifications for the CC1352P7 as an example: the current in Tx mode is 8 mA, in Rx mode it is 5.4 mA, and for the CPU it is 2.63 mA, with an operating voltage of 3.3 V. This results in a power consumption of approximately 52.9 mW (or 52.9 mJ/s).

## 6. Results

### 6.1. Linear Network

Figure 10 shows the behavior of a linear network as a function of the number of R nodes (*N*) during the association process in JS1. The solid lines with circle markers correspond to the values of $ta_{N}$ obtained via simulation for the conventional Wi-SUN FAN, while the solid lines with unfilled triangle markers correspond to those values for the PR Wi-SUN FAN. In both cases, simulations were performed for different numbers of available channels. The dotted lines with circle markers correspond to the theoretical analysis for the conventional Wi-SUN FAN, while the dotted lines with unfilled triangle markers correspond to the theoretical analysis of the PR Wi-SUN FAN, developed in Section 3 and Section 4, respectively.

Regarding the conventional Wi-SUN FAN, the results show that the association time increases with the number of channels because a joining node must hop among a greater number of channels to receive a PA. In all scenarios, this time exhibits linear growth, since each node *j* needs to wait for the connection of the previous node. The average time difference between nodes, $\Delta ta_{j}$, remains nearly constant, confirming the theoretical analysis presented in Section 3. For example, in the simulations with 90 channels (light blue solid line), $\Delta ta_{j}$ is approximately 89.7 s, a value close to $ta_{1}$, which corresponds to the first associated R node. In the case of the PR Wi-SUN FAN, it was found that, for j≥2, nodes possibly exchanged PAS packets filling the PR_table. Thus, it is likely that these nodes were connected via PA_unicast packets. For example, in the 90-channel scenario, we observe $\Delta ta_{2}=52.8$ s, $\Delta ta_{3}=28.9$ s, and $\Delta ta_{4}=23.9$ s. These early values indicate a significant improvement in association time for the PR Wi-SUN FAN in JS1 compared to the conventional Wi-SUN FAN.

It was also found that the association time in JS1 for the PR Wi-SUN FAN, compared to the Wi-SUN FAN, shows a significant reduction. For example, in the 90 CH scenario, the association time without PR was 897.4 s, while with PR it was 258 s, representing a 71.22% reduction. This resulted in a significant improvement in the time required to form the linear network.

### 6.2. Fully Connected Network

In this network, where each node can communicate directly with any other, the effect of network density was observed. Figure 11 presents a comparison between the Wi-SUN FAN and PR Wi-SUN FAN on JS1 in terms of association time when the number of R nodes varies from 10 to 50. It was observed that the association time $ta_{N}$ decreases as *N* increases in both cases. In scenarios with a greater number of available channels, the decrease in association time is more pronounced, possibly because packet collisions are avoided more frequently. Another cause can be attributed to the duration of the PA_train: the $C\cdot T_{e}$ period is longer in these scenarios, which allows more time for already associated nodes to send a greater number of PAs, thus accelerating the association of the last node. In the PR Wi-SUN FAN, there is a slight reduction in association time compared to the conventional Wi-SUN FAN, possibly because the start time of the PA_train, which controls the trickle timer, is suppressed. This occurs because the PA_unicast is sent almost immediately. Furthermore, with more channels available, nodes take longer to detect a PA packet. During this interval, they have a greater chance of receiving a PAS packet and thus begin sending PA_unicast packets, accelerating network association. For example, in the 90 CH and 50 R scenario, the association time without PR was 73.45 s, while with PR it was 57.51 s, representing a 29.87% reduction. This reflects a significant improvement in the time required to form the fully connected network.

Figure 11 also shows the comparison between the simulations and the theoretical analysis developed in Section 3 and Section 4 for Wi-SUN FAN and PR Wi-SUN FAN, respectively. In this case, it is demonstrated that the simulated results follow the trend predicted by the theoretical model. The discrepancies between the simulation results and the theoretical analysis arise from simplifications assumed in the model.

### 6.3. Mesh Network

Figure 12 shows the association time as a function of the number of nodes for a mesh topology, for both the Wi-SUN FAN and PR Wi-SUN FAN in JS1. It was observed that simulations implemented with PR exhibit a decrease in node association time, mainly due to the presence of hops within the mesh network, which allows more distant nodes a greater probability of receiving a PA_unicast packet for their association. Furthermore, it was verified that scenarios having a higher number of available channels exhibit a more significant decrease in association time. This is attributed to the longer duration of the PA_train, which gives unassociated nodes more time to receive PAS and build their PR_table. For example, in the 90 CH and 50 R scenario, the association time without PR was 117.89 s, and with PR it was 86.45 s, achieving a 26.67% reduction. This reflects a significant improvement in the mesh network formation time.

Figure 13 presents a comparison between the simulation results and the general theoretical analysis for a 90-channel mesh network, as discussed in Section 3 and Section 4, for both Wi-SUN FAN and PR Wi-SUN FAN. It can be observed that the simulation results with 90 channels fall between the upper and lower bounds predicted by the theoretical linear and fully connected network models, respectively. Therefore, the hybrid behavior of the mesh topology is confirmed.

### 6.4. Energy Considerations

Figure 14 shows the average power consumption during network formation for a linear topology with 10 R nodes, using 90 channels. The blue vertical bars represent the power consumption of the Wi-SUN FAN, and the orange vertical bars represent that of the PR Wi-SUN FAN. It can be seen that the power consumption of the nodes with the Parallel Rendezvous implementation is lower during the association process in JS1, due to the reduced association times in the PR Wi-SUN FAN.

The energy consumption of each node was calculated according to (Equation 24) to determine the value of $E_{j}$ for each node. The total energy consumption of the ten R nodes was then computed as the average of the sums of their individual energy consumptions. The results obtained were 13.16 J for Wi-SUN FAN and 5.32 J for PR Wi-SUN FAN. This indicates a 59.56% reduction in energy consumption using the proposed approach.

Figure 15 illustrates the total energy consumption for fully connected and mesh networks with 10, 20, 30, 40, and 50 nodes, operating over 90 channels. The result shows that the energy consumption during the association process using the PR method is consistently lower compared to the conventional approach.

The total energy consumption was computed as the average sum of the consumption across all nodes. For instance, in a fully connected network having 50 nodes, the energy consumption was 1.19 J for Wi-SUN FAN and 0.75 J for PR Wi-SUN FAN. In the mesh network, the consumption was 1.43 J for Wi-SUN FAN and 0.94 J for PR Wi-SUN FAN. These values represent a 37% reduction in the fully connected network and a 34.3% reduction in the mesh network, demonstrating a significant decrease in energy consumption during the association process in JS1.

## 7. Conclusions

In this work, we evaluated the node association process in JS1 of Wi-SUN FAN networks through theoretical analysis and simulations. It was demonstrated that the association process time in JS1 can be reduced by adopting the concept of Parallel Rendezvous, for which a new algorithm, PR Wi-SUN FAN, was proposed. The performance of our proposal was compared with the original version of the standard, based on the average association time, showing improved performance in linear, fully connected, and mesh topologies, with the best results observed in linear networks, achieving up to a 71.22% reduction in association time in JS1. Energy measurements indicated that the implementation introduces significant changes for the R nodes, achieving a reduction of up to 59.56% in energy consumption in JS1 in linear topology.

Future work will involve implementing and evaluating our proposal alongside the other joining states to comprehensively assess the performance of PR Wi-SUN FAN throughout the entire connection process, using both simulations and real-world experiments. This evaluation will also consider another type of channel function with a different focus on sequence length. Other scenarios, such as the disconnection or reset of operating nodes, can also be evaluated.

## Figures and Tables

**Figure 1 sensors-25-06213-f001:**
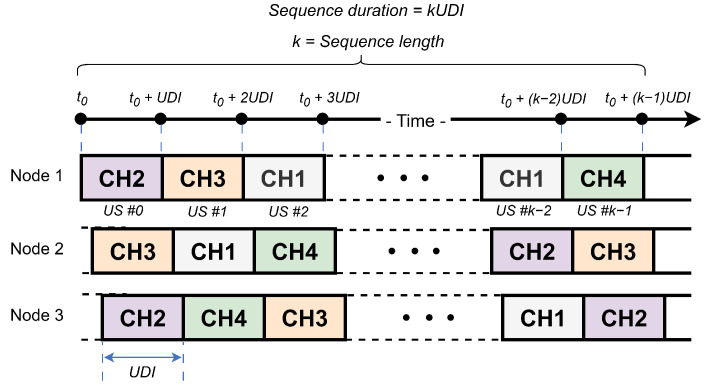
Example of Wi-SUN FAN scheduling.

**Figure 2 sensors-25-06213-f002:**
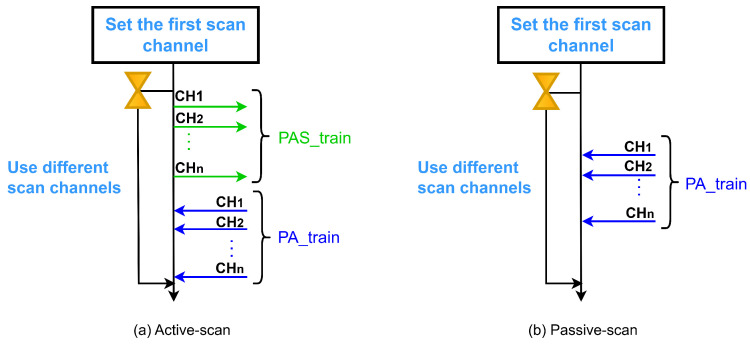
Active and passive scanning process.

**Figure 3 sensors-25-06213-f003:**
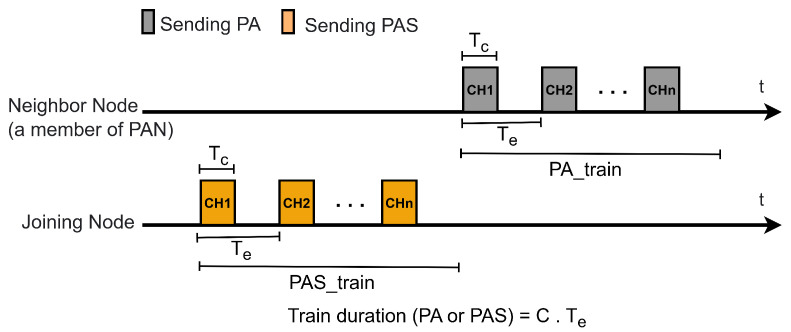
Packet train transmissions.

**Figure 4 sensors-25-06213-f004:**
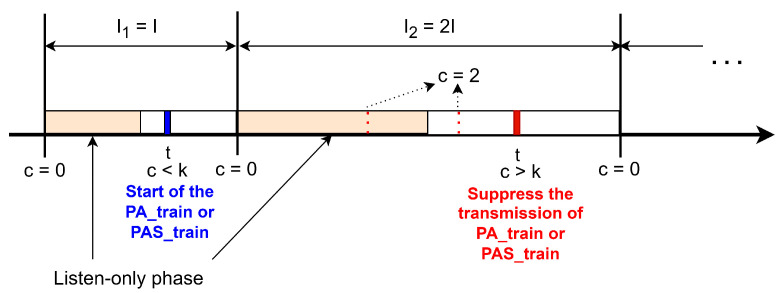
Two-interval trickle timer with k=1, where $I_{1}$ denotes the initial interval and $I_{2}$ the interval obtained by doubling the previous one. The blue line represents a transmission—in Wi-SUN FAN, this corresponds to the start of the PA_train or PAS_train. The red line denotes a suppressed transmission, caused by the reception of a PA or PAS packet, which is indicated by the red dotted lines.

**Figure 5 sensors-25-06213-f005:**
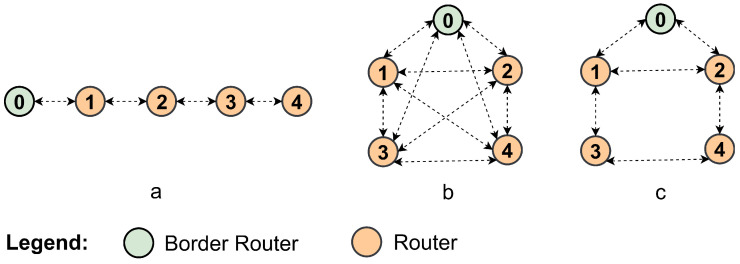
Network topology examples: (**a**) linear, (**b**) fully connected, (**c**) mesh.

**Figure 6 sensors-25-06213-f006:**
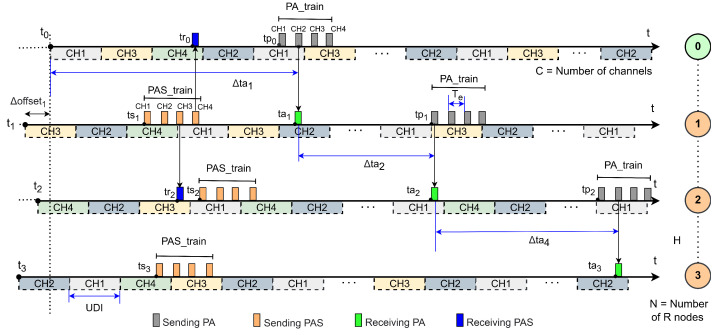
Example of the node association process in JS1 in a linear topology in Wi-SUN FAN networks.

**Figure 7 sensors-25-06213-f007:**
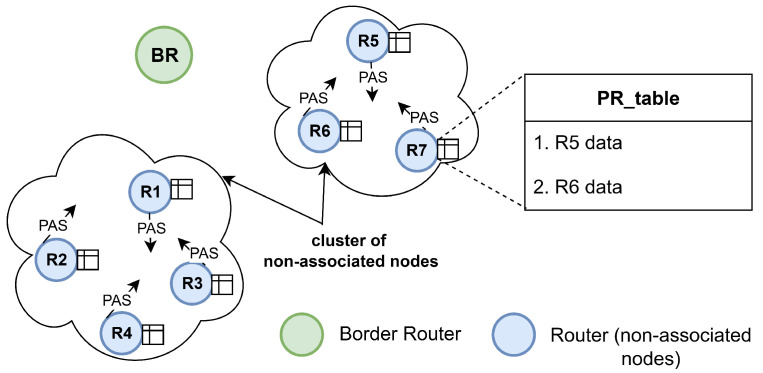
Example of cluster formation in PR Wi-SUN FAN.

**Figure 8 sensors-25-06213-f008:**
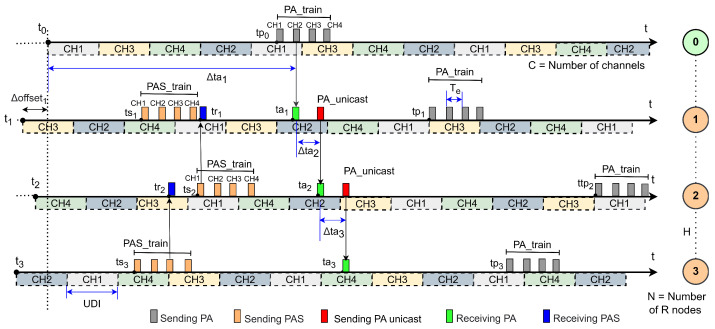
Example of the node association process in JS1 in a linear topology in PR Wi-SUN FAN networks.

**Figure 9 sensors-25-06213-f009:**
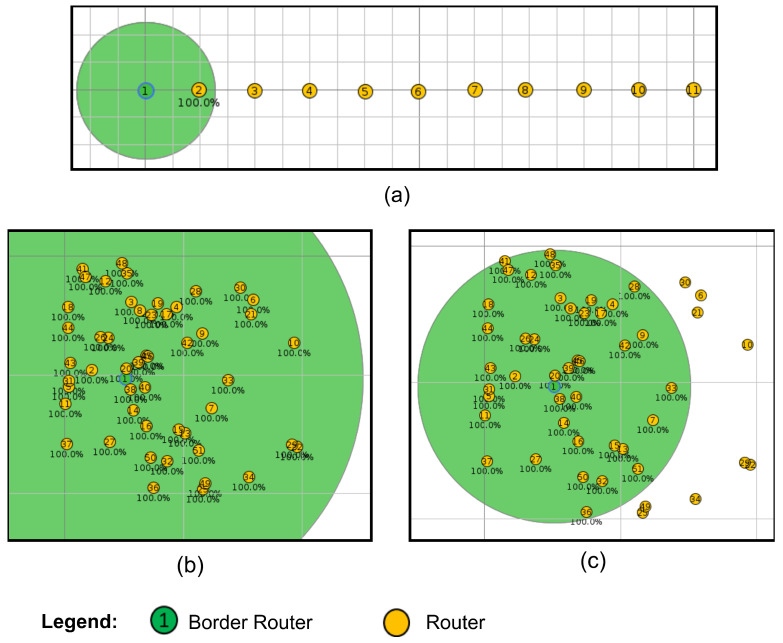
Topologies in Cooja: (**a**) linear; (**b**) fully connected; (**c**) mesh.

**Figure 10 sensors-25-06213-f010:**
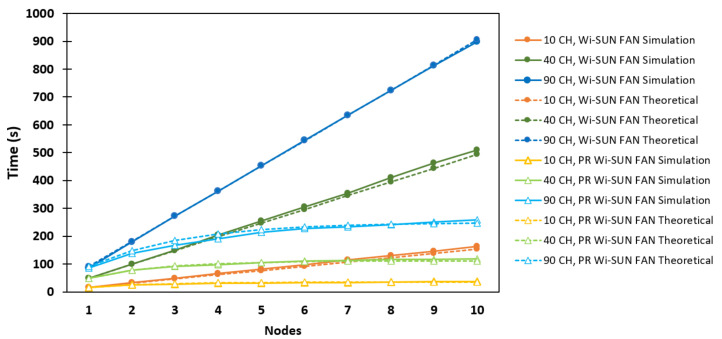
Comparison between Wi-SUN FAN and PR Wi-SUN FAN simulation algorithms in a linear topology.

**Figure 11 sensors-25-06213-f011:**
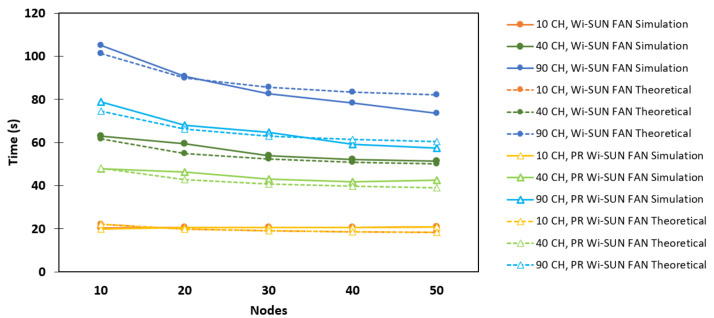
Comparison between Wi-SUN FAN and PR Wi-SUN FAN simulation algorithms in a fully connected topology.

**Figure 12 sensors-25-06213-f012:**
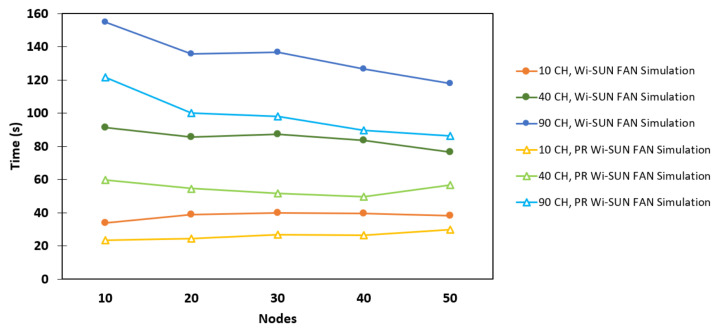
Comparison between Wi-SUN FAN and PR Wi-SUN FAN simulation algorithms in a mesh topology.

**Figure 13 sensors-25-06213-f013:**
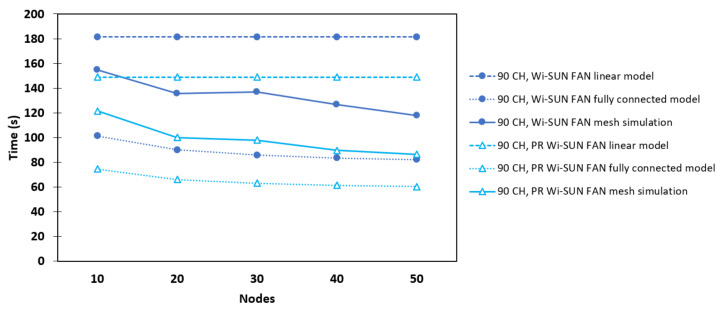
Simulation results for the Wi-SUN FAN and PR Wi-SUN FAN networks as a function of the number of nodes in the mesh network (solid lines with circle and triangle markers, respectively) compared to the theoretical models for linear (dashed lines with circle and triangle markers, respectively) and fully connected (dotted lines with circle and triangle markers, respectively) networks, using 90 channels.

**Figure 14 sensors-25-06213-f014:**
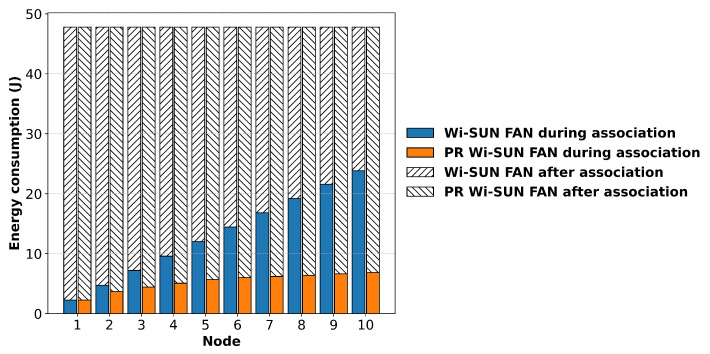
Average power consumption of Wi-SUN FAN and PR Wi-SUN FAN in a linear network.

**Figure 15 sensors-25-06213-f015:**
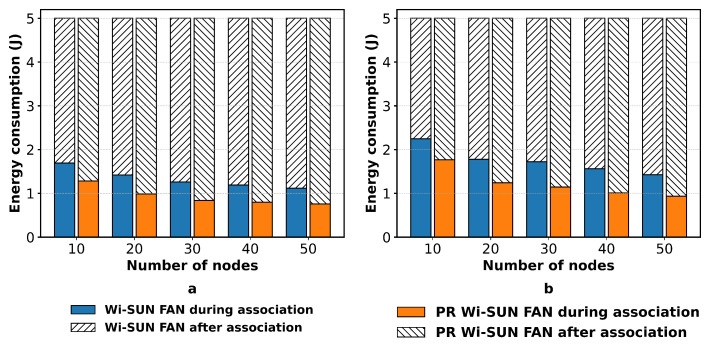
Average power consumption of Wi-SUN FAN and PR Wi-SUN FAN: (**a**) fully connected; (**b**) mesh.

**Table 1 sensors-25-06213-t001:** Configurations used for different numbers of channels.

Number of Channels	UDI (ms)	$T_{e}$ (s)
10	100	1
40	50	2
90	20	1.8

## Data Availability

Data are contained within the article.

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
