# Peer review of "Parallel Rendezvous Strategy for Node Association in Wi-SUN FAN Networks"

_sensors, 2025, doi:10.3390/s25196213_

Round 1
Reviewer 1 Report
Comments and Suggestions for Authors
Please see the pdf file.

Author Response
Dear,
I have attached a document with the answer.

Reviewer 2 Report
Comments and Suggestions for Authors
The manuscript presents two algorithms for parallel Rendezvous strategy for node association in Wi-SUN FAN Networks based on a previous work from the authors [33]. In particular, a conceptual analysis of the joint state 1, i.e., the PAN (Personal Area Network) discovery and selection state, is presented as one of the main contributions. However, the manuscript suffers from several shortcomings that significantly limit its contribution and impact. In particular, authors must deal with the following issues in order to improve the overall presentation of the manuscript.
1) The introduction lacks clarity in defining the research gap, the novelty of the proposed strategy is not sufficiently demonstrated, and the practical application to smart grids is underdeveloped. Only two references mention the smart grid problem, but it is not directly stated. The mention of the smart grid application in the title seems artificial, since it is not developed in the rest of the paper. We expect particular simulations in the context of smart grids, and a more elaborated context of the particular problem. The introduction does not explicitly articulate the core contribution of the paper. It is difficult for the reviewer to identify what is fundamentally new about the proposed parallel rendezvous strategy in comparison with previous work. Not up-to-date recent literature is cited in the introduction, only one paper from 2025. While the context of Wi-SUN FAN networks is outlined, the gap in existing literature is not adequately highlighted. The paper should clearly state how this strategy advances beyond previous approaches in node association and in particular the benefits in smart grids applications. Although the title emphasizes “smart grid application,” the paper does not sufficiently demonstrate how the proposed method addresses the unique challenges of smart grid communication networks. The validation examples are generic and do not convincingly reflect realistic smart grid conditions such as scalability, reliability under stress, or interoperability with existing standards. The paper would benefit from a stronger case study or simulation framework explicitly tied to smart grid requirements.
2) The strategy appears to be an incremental modification of existing rendezvous mechanisms, without a clear theoretical or practical application in smart grids. The results, as presented, do not convincingly show that the proposed approach significantly improves network performance compared to established methods. The lack of comparison with state-of-the-art strategies further weakens the claimed contribution.
3) More rigorous performance evaluation in the simulation section is required, ideally including benchmarks against standard node association techniques in Wi-SUN FAN networks. The implications of the proposed strategy for real-world deployment in smart grid systems should be discussed in greater depth.
Author Response

(The authors gave the same response as above.)

Reviewer 3 Report
Comments and Suggestions for Authors
please see the attached pdf file

Author Response

(The authors gave the same response as above.)

Reviewer 4 Report
Comments and Suggestions for Authors
The growing demand for connectivity and low-cost solutions has led to the widespread adoption of low-data-rate, low-power communications, especially important in Smart Grid applications for efficient monitoring and control. The Wi-SUN FAN (Wireless Smart Ubiquitous Network Field Area Network) standard is notable for enabling large-scale connections among smart devices in utility networks and smart cities. Designed for Low Power and Lossy Network (LLNs), Wi-SUN FAN supports the formation of multiple Personal Area Networks (PANs) and mesh topologies with multi-hop transmissions and channel hopping, using open protocols compatible with IEEE 802.15.4. However, the node association process, which is divided into five junction states, often results in prolonged connection times, particularly in multi-hop networks, which undermines network scalability and reliability-key requirement for Smart Grid field-area networks. This study analyzes the factors influencing these periods, with particular emphasis on the Join State 1 (JS1), which uses asynchronous communication and the trickle timer algorithm to schedule PAN Advertisement (PA) packets. Depending on the standard configuration, this can lead to extended transmission intervals. To address this, an approach based on forming synchronized clusters among unassociated nodes, leveraging PAN Advertisement Solicit (PAS) packets and the parallel rendezvous (PR) strategy to rapidly disseminate network information was proposed. The proposed algorithm, PR Wi-SUN FAN, is evaluated through simulations across various network topologies, demonstrating better results for a linear network, with reductions of up to 71.22% in association time and 59.56% in energy consumption during JS1.
The topic is interesting, and the text is well-structured. However, the following main problems in this paper need to be solved.
- The methods adopted in the manuscript lack sufficient novelty and are not suitable for publication in a journal with an impact factor like Sensors.
- It is difficult to prove the effectiveness of the method proposed by the author, so please supplement the corresponding real experiments to the simulation experiment section.
Author Response

(The authors gave the same response as above.)

Round 2
Reviewer 1 Report
Comments and Suggestions for Authors
No further comments.
Reviewer 2 Report
Comments and Suggestions for Authors
The authors have properly responded to my concerns and properly included the suggestions.
Reviewer 3 Report
Comments and Suggestions for Authors
The paper can now be accepted.
Reviewer 4 Report
Comments and Suggestions for Authors
The author has made some modifications, it's great! However, some places should be further improved, for example,
As the authors said in their reply to the peer reviewers' comments, this paper has no corresponding real experiments to the simulation experiment section. However, papers without real experiments are not suitable for publication in sensor journals with impact factors.